# Novel approach to estimate tuberculosis transmission in primary care clinics in sub-Saharan Africa: protocol of a prospective study

Kathrin Zürcher ![ORCID],[1] Carl Morrow,[2,3] Julien Riou,[1] Marie Ballif ![ORCID] ,[1] Anastasia Sideris Koch ![ORCID] ,[3,4] Simon Bertschinger,[1,5] Xin Liu,[6] Manuja Sharma,[6] Keren Middelkoop ![ORCID] ,[2,3] Digby Warner,[3,4] Robin Wood,[2,3] Matthias Egger ![ORCID] ,[1,7,8] Lukas Fenner ![ORCID] [1]

For numbered affiliations see end of article.

**Correspondence to**
Dr Lukas Fenner;
lukas.fenner@ispm.unibe.ch

## ABSTRACT

**Introduction** Tuberculosis (TB) transmission is difficult to measure, and its drivers are not well understood. The effectiveness of infection control measures at healthcare clinics and the most appropriate intervention strategies to interrupt transmission are unclear. We propose a novel approach using clinical, environmental and position-tracking data to study the risk of TB transmission at primary care clinics in TB and HIV high burden settings in sub-Saharan Africa.

**Methods and analysis** We describe a novel and rapid study design to assess risk factors for airborne TB transmission at primary care clinics in high-burden settings. The study protocol combines a range of different measurements. We will collect anonymous data on the number of patients, waiting times and patient movements using video sensors. Also, we will collect acoustic sound recordings to determine the frequency and intensity of coughing. Environmental data will include indoor carbon dioxide levels ($CO_2$ in parts per million) and relative humidity. We will also extract routinely collected clinical data from the clinic records. The number of *Mycobacterium tuberculosis* particles in the air will be ascertained from dried filter units using highly sensitive digital droplet PCR. We will calculate rebreathed air volume based on people density and $CO_2$ levels and develop a mathematical model to estimate the risk of TB transmission. The mathematical model can then be used to estimate the effect of possible interventions such as separating patient flows or improving ventilation in reducing transmission. The feasibility of our approach was recently demonstrated in a pilot study in a primary care clinic in Cape Town, South Africa.

**Ethics and dissemination** The study was approved by the University of Cape Town (HREC/REF no. 228/2019), the City of Cape Town (ID-8139) and the Ethics Committee of the Canton Bern (2019-02131), Switzerland. The results will be disseminated in international peer-reviewed journals.

## Strengths and limitations of this study

► We describe the protocol for a prospective study design to studying tuberculosis (TB) transmission in primary care clinics in high TB/HIV-burden settings.

► This rapid approach will combine a wide range of different measurements, including patient waiting times and movements, acoustic recording of coughing, measurement of carbon dioxide levels as a natural tracer gas, air humidity and semiquantitative detection of *Mycobacterium tuberculosis* (Mtb) particles in the air.

► We will develop a mathematical model which will integrate the collected data to estimate the risk of TB transmission, identify key drivers of transmission and evaluate the impact of infection control measures such as improved ventilation or wearing masks.

► The main limitation of this study design is the lack of direct observation of transmission events and the reliance on the number of Mtb particles in the air as a proxy for TB transmission.

► Study limitations pertain to the need for stable electricity and WiFi for data collection.

## INTRODUCTION

Tuberculosis (TB), caused by the bacterium *Mycobacterium tuberculosis* (Mtb), remains a major global public health problem, particularly in the context of HIV and drug resistance. Sub-Saharan Africa is one of the most heavily burdened regions globally, although control measures have been in place since the beginning of the 20th century.[1] Over a century of investment in TB control has reduced TB mortality, but effective strategies are urgently needed to reduce TB transmission.[2] Drivers of the TB epidemic in sub-Saharan Africa are HIV-infection and the resulting immunodeficiency (the strongest risk factor),[1] delayed diagnosis and treatment as well as undetected and untreated cases of TB or drug-resistant TB. These factors allow patients with infectious TB to transmit Mtb to the community.[3–5]

There are still many gaps in our knowledge on TB transmission such as the factors and locations associated with the risk of transmission, the effectiveness of infection control measures at clinics in high-burden settings and the most appropriate intervention strategies to interrupt transmission.[5–7]

For TB transmission to occur, infected individuals must expel Mtb bacilli from their respiratory tract, and an uninfected individual must inhale aerosols containing live bacilli to become infected. Transmission of Mtb is difficult to measure due to the lack of an in vitro test assay. The preferred approach is to measure presumptive transmission resulting in secondary cases as determined by molecular/genomic epidemiology. TB transmission has traditionally been investigated using contact tracing, analyses of geo-temporal clustering and molecular typing.[8 9] However, molecular methods require resource-intensive culturing of strains and measure only transmission resulting in secondary cases. Furthermore, contact tracing is difficult to implement in resource-limited settings and may not be an effective control strategy in endemic areas where casual contacts are increasingly recognised as an important contributor to transmission.[10] New approaches are therefore urgently needed.

Mtb is carried in airborne particles (called infectious droplets), which are generated when people with TB cough, sneeze or shout.[5] Indoor carbon dioxide ($CO_2$) levels can be used to assess the amount of exhaled air in a room and the amount of rebreathed air.[11–13] Humidity is associated with Mtb survival in the air.[14] Viable Mtb particles have been captured from contaminated air.[11 15–17] We describe a unique study design to assess risk factors for airborne TB transmission in primary care clinics in high TB/HIV-burden settings. The approach combines a range of relevant measurements, including data on patient and infrastructure, movements of patients through the facility, coughing, environmental indoor $CO_2$ levels and concentration of Mtb particles in the air.

We hypothesise that (1) exposure to Mtb particles at the clinic can be estimated by studying the patient flow and $CO_2$ levels; (2) the number of individuals present, the rebreathed fraction and the frequency of coughs influence exposure to Mtb; (3) clinics with small and crowded waiting rooms, low ventilation and suboptimal patient separation have an increased risk of TB transmission.

## METHODS
### Study design
Longitudinal study with data collection at the levels of the patients, the clinic and the environment. Data collection will take place for over 4 weeks. Figure 1 shows the floor plan of an exemplary primary clinic with the planned study activities, including the recording of patient movements, coughing, $CO_2$ levels and Mtb particles in the air.

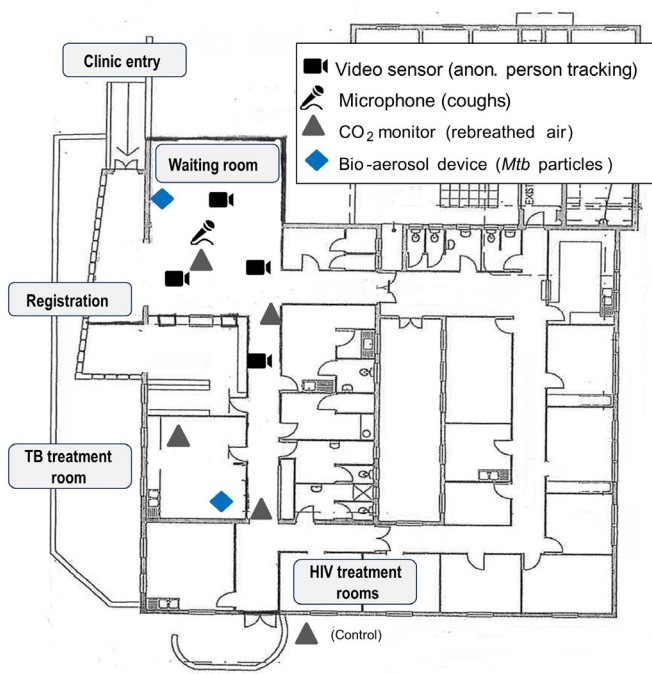

**Figure 1** Floor plan of the pilot primary care clinic in Cape town, South Africa, indicating study measurements. Mtb, *Mycobacterium tuberculosis*; TB, Tuberculosis.

### Study setting and study population
The study will take place at several antiretroviral therapy (ART) or primary care clinics in countries participating in the International Epidemiology Database to Evaluate AIDS (IeDEA) in Southern Africa collaboration with a high TB and HIV burden.[18] These clinics are located in urban and peri-urban communities with predominantly young and black African residents. TB and HIV are both prevalent in these communities.

### Patient-level variables
#### Video sensor data
A person-tracking sensor system developed by Xovis (Zollikofen, Switzerland and Cambridge, Massachusetts, USA; see www.xovis.com) will be used to monitor the clinic attendees' movements. The data will be used to calculate waiting times, the number of people in different locations and the average distance between people, and to identify highly frequented areas. Several sensors will be installed to cover the clinic area, calibrated and validated. Sensors with overlapping ranges will be combined for the seamless coverage of people's movements over large areas (figure 2). The raw data consist of the person's height, time, date and the position (x–y coordinates) for each unique individual during the duration of their stay within the clinic. The data are captured every 0.25 s. The raw data are then parsed by a python script to calculate the height, total movement and observation time at different locations, and to visualise hotspots.

The sensors have four levels of privacy. In our study, privacy will be set to level 2, which means that the data are a fully anonymised stream of the coordinates of

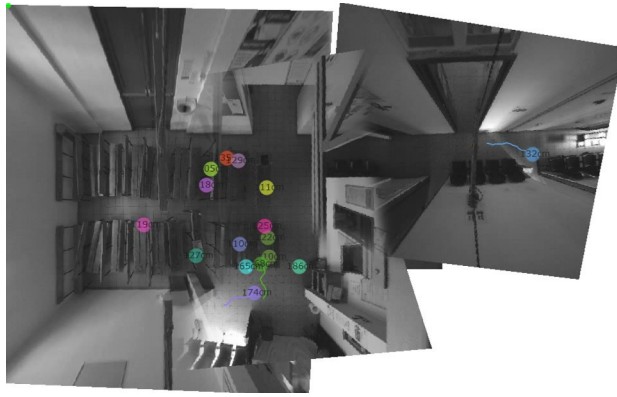

**Figure 2** Output from video sensors with moving dots showing the tracked persons. The numbers in the dots indicate the height of patients. The different sectors covered by the sensors are merged images from the pilot study.

moving dots. These dots will be linked probabilistically to the anonymised clinical data thus excluding any risk of re-identification. Further, the images of tracked individuals taken by the sensors will not be stored (see also the data privacy and security statement by Xovis).[19 20]

### Patient and clinic data

We will extract clinical data from the electronic patient registry for all patients who visited the clinics during the 4 weeks of data collection. The data will include date and time of registration, age, sex, height (used for linkage with video sensor data), HIV status, presumptive TB ('TB suspects'), mode of TB diagnosis (sputum smear microscopy, Xpert MTB/RIF, mycobacterium culture), TB diagnostic test results, date of TB treatment start and current anti-TB treatment regimen. We will not extract any personal data, such as names or social security numbers. The clinic-level data will include information such as the setting (urban or rural), level of care, the number of patients seen each year, availability of adult and paediatric care, TB control measures (natural ventilation, use of masks, separation of patients with TB or coughers) and the floor plan of the clinics. The data will be collected using the web-based REDCap Data Entry System (www.redcap.org).[21]

### Definitions

Presumptive TB refers to a patient who presents with clinical symptoms or signs suggestive of TB. A bacteriologically confirmed TB case is a person for whom a biological specimen is positive by smear, microscopy, culture or rapid diagnostic such as Xpert MTB/RIF or line probe assay.[22]

### Environmental variables

#### $CO_2$, relative humidity and temperature monitoring

We will use $CO_2$ monitors (Digital $CO_2$ Monitor Carbon Dioxide Metre XE-2000, XEAST, Guangdong, China) which include the COZIR-A sensor for ambient $CO_2$ levels of 0%–1% (Gas Sensing Solutions, Cumbernauld, Scotland). The monitors will record indoor $CO_2$ concentrations (in parts per million (ppm)), temperature and relative humidity at minute intervals. Data from the five monitors are stored and exported in serial digital format.[15 16] At each clinic we will install five monitors to cover the most visited spaces and a control area outside the clinic (figure 1). The monitors auto-calibrate over time to a standard minimum value of 400 ppm. This value is close to the average monthly outdoor carbon dioxide concentration measured at Cape Point, South Africa (405.48 ppm in December 2018).[23] Monitors will be run for 1 week before the start of the study to allow them to calibrate and to confirm that they delivered comparable data.

### Cough monitoring

We will install a microphone (RØDE NT-USB, Sydney, Australia) near the ceiling to continuously record the sounds in the waiting room (figure 1). CoughSense, a deep learning cough detection algorithm based on MXNet, an open-source deep-learning software framework, was developed to classify audio signals as coughing or other sounds.[24] The algorithm uses spectrograms extracted from the raw audio for classification. The model was trained and tested using multiple audio recordings obtained through clinical and ambulatory deployments. We will identify audio records with cough sounds and calculate the cough frequency, intensity and duration. The cough data will be linked with the $CO_2$ data and video sensor data by time and date. Finally, we will probabilistically link the video sensor data (ie, the moving dots) with the clinical data by the height of the patient, time and date of the visit.

### Bio-aerosol sampling and molecular detection

We will collect Mtb particles from the air using mobile bio-aerosol sampling devices (Dry Filter Unit (DFU) 1000, Lockheed Martin Integrated Systems, Gaithersburg, Maryland, USA). The DFU 1000 is a portable biological air sampler. Ambient air is drawn through 1 μm polyester-felt filters at a rate of ~1000 L/min via an electrical blower. The DFU is a useful collection system for low concentration aerosols, capturing particles of 1 μm or larger, and allowing easy access for the retrieval of filters. One DFU will be placed in the waiting room and the other in the TB treatment room (figure 1). Each DFU collects air for 7 hours (two periods of 3.5 hours) every day onto two filters. Due to logistical and human resource limitations, we will be able to change filters only two times per day. Therefore, we decided to run the mobile bio-aerosol sampling devices during the busiest time at the clinic, between 07:00 and 14:00 . All other data will be collected from 07:00 to 16:00 .

As previously described,[17] duplicate filters from each sampling session will be transferred to 50 mL Falcon tubes and vortexed in sterile phosphate buffered saline (with 0.05% Tween 80). Following centrifugation at 3750 rpm for 15 min, filters will be removed and the pellet subjected to DNA extraction. DNA from Mtb cells will

**Table 1** Description of the measurements

| Data source | Parameter | Description | Unit | Measurement taken by |
|---|---|---|---|---|
| $CO_2$ monitor | $CO_2$ | Observed $CO_2$ concentration in the indoor air per minute and a control in the outdoor air. Based on $CO_2$ levels and people density, we will calculate rebreathed air volume, which is used as a proxy for airborne TB transmission.[16] | ppm | Minute and date |
| | Relative humidity | Data on the effects of relative humidity on the survival of airborne bacteria are inconsistent.[28 37] However, a recent study found that relative humidity above 65% is associated with Mtb survival in the air.[14] | % | Minute and date |
| | Temperature | Temperatures above 24°C are required to reduce airborne bacteria survival.[28 37] | °C | Minute and date |
| Cough recording | Frequency | One of the typical symptoms of TB is coughing; coughing is also the main way of transmission. | n | Minute or day and date |
| | Duration | Duration of each cough is different from healthy and people with TB or other lung diseases[30] | s | Cough by minute and date |
| | Intensity | Intensity of each cough is different from healthy and people with TB or other lung diseases.[30] | dB | Cough by minute and date |
| Mobile aerosol sampling | Mtb DNA copies | Detection of Mtb particles in the air by filter or per day (07:00 to 14:00).[17] | Copies per microliters | Filter (ca.3.5 hours sampling) or per day |
| Video sensor | Number of people | From the raw data (x–y coordinates) we can calculate the number of people at a given location by 0.25 second and by minute. | n of people | 0.25 s or min and by date |
| | Time spent at a given location | From the raw data (x–y coordinates) we can calculate for each person their time spent at different locations. | min | Minute and date |
| Patient charts | Number of registered patients | All patients who are visiting the clinic are registered. | n of registered patient | Minute and day |
| | Number of presumptive TB and of TB patients | From all registered patients we will know the number of presumptive TB cases and the number of patients with TB. | n of presumptive TB and TB patients | Minute and day |

dB, decibel; Mtb, *Mycobacterium tuberculosis*; n, number; ppm, parts per million; s, second; TB, tuberculosis.

be extracted using an in-house lysis buffer with subsequent pelleting (centrifugation 13 000 rpm for 10 min) of DNA and resuspension in 50 µl of Tris-EDTA buffer (10 mM Trist, 1 mM EDTA, pH 8.0). Given that droplet digital PCR (ddPCR) is relatively robust against inhibitors, no further DNA purification will be required. The primer/probe combinations and reaction conditions for Mtb-specific ddPCR have been described.[16] Samples with known amounts of purified Mtb DNA (0.01 ng and 0.001 ng) will be included to serve as positive and nuclease-free water as negative control. The data generated from the ddPCR reaction will be analysed via the Umbrella pipeline,[25] using wells with a minimum of 10 000 droplets.

## Statistical analyses

We will describe the data and examine associations between sources of data and then use results to parameterise a mathematical model (table 1). The statistical analysis will quantify the joint association between the clinical and environmental variables and the number of Mtb particles measured by the mobile aerosol sampling. To this end, we will use Poisson regression, with the number of Mtb particles by the period of time as the dependent variable, considered as a proxy measure for the risk of TB transmission. As this variable is measured by periods of

3.5 hours, the other variables will be aggregated over the same periods. We will consider all combinations of independent variables, including second-order interactions, and compare the model using standard model selection methods. We will use variable selection methods in a Bayesian framework, including the deviance information criterion and the leave-one-out information criterion.[26]

We will operationalise the variables as follows:

1. *Patient data*: Numbers and characteristics of patients consulting the clinic overall will be summarised using descriptive statistics.
2. *Video sensor data:* Raw data about individuals' movements will be transformed into waiting times until medical consultation, number of individuals in the different locations, highly frequented areas in the clinic and the average distance between patients (ie, clustering). We will link each tracked individual to the clinical data collected from the clinic's database (ie, TB/HIV diagnosis), using the order of arrival and time of registration.
3. *$CO_2$ data*: Measurements of $CO_2$ concentration at the different locations in the clinic (waiting room, registration desk, TB treatment room), together with estimates in outdoor air, will be used to estimate the proportion

of air in the different locations that was expired by individuals, the rebreathed fraction.[12] We will use a modified Wells-Riley model appropriate for non-steady states conditions of ventilation and number of individuals to describe and calculate the shared rebreathed air in the different locations in either litre/minute or litre/day and the air exchange (litre/hour per person). Table 2 describes the parameters to calculate the shared rebreathed air and the air exchange.[13 16 27 28]

The proportion of rebreathed air ($f$) will be calculated from the excess $CO_2$ measured indoors, divided by the exhaled $CO_2$ ($Ca$):

$$f = \frac{(C-Co)}{Ca} \qquad (1)$$

(see table 2 for definitions of parameters in this and the following equations).

Before we can calculate the rebreathed air volume (RAV) we will need the rebreathed proportion from other people ($fo$). Therefore, we need to know the number of people ($n$) present at each time point at the clinic:

$$fo = f \times \frac{(n-1)}{n} \qquad (2)$$

Finally, we can calculate the RAV for each minute by multiplying $fo$ and the minute respiratory volume (8 L/min, ($p$)) as:

$$\text{Rebreathed air volume} \left(\text{RAV}\right) = \left(pfo\right) \qquad (3)$$

In the next step, we aim to calculate the ventilation rate (air exchange). The indoor $CO_2$ generated rate is the product of the average volume of gas exhaled per person (0.13 L/s per person, ($V$)) and the $CO_2$ concentration of the exhaled air:

$$\text{Indoor } CO_2 \text{ generation rate in l/s}$$
$$\text{per person} \left(G\right) = V x Ca \qquad (4)$$

The ventilation rate is expressed as:

$$\text{Ventilation rate in l/s per person}$$
$$\left(Q\right) = \frac{G}{C-Co} \qquad (5)$$

To calculate the air exchange per hour (Equation 6) we need to know the volume of air in a given location.

4. *Relative humidity and temperature:* We will describe changes in relative humidity and temperature over the day.
5. *Cough sounds:* Frequency, intensity, duration of recorded coughs per time period.[29 30]
6. *Detection of Mtb in bio-aerosol sampling:* The number of Mtb genome copies present in each sample.

**Table 2** Description of the variables to calculate the shared rebreathed air volume and air exchange as well as the parameters to construct the mathematical transmission model

| Parameter | Description | Value |
|---|---|---|
| C | Observed $CO_2$ concentration in the indoor air per minute | Observed |
| Co | $CO_2$ concentration in the outdoor air per minute | 400–420 ppm[23] |
| Ca | $CO_2$ concentration in the exhaled air | 38 000–40 000 ppm[13 38] |
| f | Proportion of rebreathed air | Equation |
| n | Number of people recorded at the location | Observed |
| fo | Rebreathed proportion from other people | Equation |
| p | Minute respiratory volume | 8 L/min[38] |
| RAV | Rebreathed air volume | Equation |
| V | Average volume of gas exhaled per person | 0.13 L/s per person[16] |
| G | Indoor $CO_2$ generation rate (L/s per person) | Equation |
| Q | Ventilation rate (L/s per person) | Equation |
| vol | Volume of the room | Calculated |

## Mathematical modelling

A mathematical model (figure 3) will integrate all sources of data to model the risk of TB transmission and identify key drivers of transmission.[13 27 31 32] We hypothesise that the number of individuals present, the rebreathed fraction and the frequency of coughs will have the greatest influence on the risk. As we do not observe transmission events, we will use $yu$, the number of Mtb genomes counted by the bio-aerosol sampling for each 6 hour period $u$, as the dependent variable. The model will describe $yu$ as a Poisson process with time-dependent intensity $\lambda t$ (also called a Cox process). This intensity can be interpreted as a proxy for the (unobserved) risk of TB transmission. We will model $\lambda tu$ using 10 min time periods $t$ within $u$. In a first simple model, $\lambda tu$ will integrate multiple independent variables $xtu$: the number of individuals in the waiting room during time $t$, the rebreathed fraction and

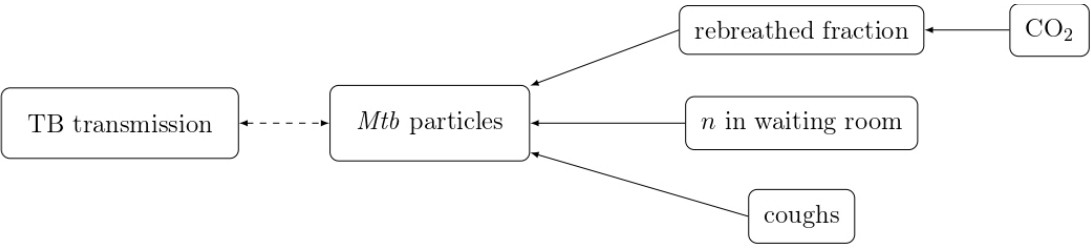

**Figure 3** Model structure. Mtb, *Mycobacterium tuberculosis*; TB, tuberculosis

the number of coughs. The model will be estimated in a Bayesian framework using Stan, a probabilistic programming language.[33]

We will use several metrics to assess the goodness of fit of the model. We will then add more complexity to the model by integrating other independent variables such as the characteristics of the patients, shared rebreathed air ($CO_2$ monitors),[12 13 15 16] patient flow (clustering of individuals, movements of patients) and other potential drivers (intensity/frequency/duration of coughs).[29 30]

Having developed the model of the risk of TB transmission, we will evaluate the possible effect of interventions such as limiting the number of patients in the waiting room, separating coughers or increasing ventilation. All analyses will be performed in R (V.3.6.0) or Stata (V.15.1, Stata Corporation, Texas, USA).

### Reporting

The results from this study will be reported following the recommendations of the Strengthening the Reporting of Observational Studies in Epidemiology statement.[34]

### Pilot study

We conducted a pilot study in Cape Town (South Africa) to examine the feasibility of our approach (figure 1). The pilot study took place at a primary care clinic, which incorporates HIV counselling and testing, and a TB clinic for the diagnosis and management of TB. The clinic is situated in Masiphumelele, a large settlement of formal and semiformal housing in Cape Town, which has previously been described.[35 36] The clinic is open on workdays from Monday till Friday from 07:00 till 16:00. We studied clinic activities for over 4 weeks on workdays between July 25 and August 23 2019.

Data collection was successful overall, however, we experienced several power cuts during the pilot study. An important lesson learnt is that power banks for laptops and WiFi routers are needed, as well as Uninterruptible Power Supply (UPS) and access to a generator as back-up for the DFU and $CO_2$ monitors.

### Patient and public involvement

We discussed the aims and study design with local clinic staff, colleagues at the University of Cape Town and public health specialists early on in the planning phase and developed the specific objectives and data collection procedures in collaboration with them. Patients were not involved in the design, recruitment or conduct of the study. We will make the results of this study available to the participating clinics and the public health authorities.

### DISCUSSION

TB control is particularly relevant for sub-Saharan Africa, which carries a disproportionally large portion of the global burden of both TB and HIV. There is an urgent need to understand the drivers of TB transmission to reduce TB incidence using new intervention approaches.[1]

A better understanding of transmission, coupled with a rapid test system are likely to contribute to improving TB control in clinic settings. This project will provide new insights into the complex TB transmission framework at a primary care clinic in an endemic setting using clinical data, $CO_2$ levels, cough analyses, video tracking and Mtb particles sampled from the air. Although the large-scale implementation and evaluation of interventions are beyond the scope of this study, the results will generate new hypotheses and opportunities for public health intervention studies (eg, randomised controlled or cluster-randomised trials).

Importantly, with mathematical models based on real-life data, we can evaluate the likely effect of interventions thus improving intervention studies and inform help with logistical and infrastructural planning of primary care clinics to reduce the transmission risk. Having established feasibility in one clinic in South Africa, we are planning to collect data in several countries in sub-Saharan Africa. The results from this broader study have the potential to inform national and international guidelines to reduce TB transmission at healthcare centres.

### Strengths and limitations

This is a novel and rapid approach to studying TB transmission combining a wide range of different measurements, which goes beyond the traditional methods such as contact tracing, geo-temporal clustering or molecular genotyping. It will lead to a comprehensive transmission model to measure the effects of various interventions in the clinic setting, paving the way for future studies. Study limitations pertain to the need for stable electrical power and WiFi for over 24 hours for data collection at the primary care clinic, which is often an issue in low-income and middle-income countries. We will address this problem by using UPS power stabilisers with access to generator back-up power.

### ETHICS AND DISSEMINATION

The University of Cape Town Faculty of Health Sciences Human Research Ethics Committee (HREC/REF: 228/2019), the City of Cape Town (Project ID 8139) and the Ethics Committee of the Canton of Bern (2019-02131), Switzerland approved the pilot study. Expansion of the study will include other clinics in the greater Cape Town metropolitan area, but also clinics in Zambia, Zimbabwe or Malawi, including urban and rural sites to ensure representative and generalisable results. Separate ethical approval will be sought from the relevant ethics committees or institutional review boards.

The results will be disseminated in international peer-reviewed journals and presented at national and international conferences.

**Author affiliations**
[1]Institute of Social and Preventive Medicine (ISPM), University of Bern, Bern, Switzerland

[2]Desmond Tutu HIV Centre, Institute of Infectious Disease and Molecular Medicine (IDM), University of Cape Town, Cape Town, South Africa
[3]Institute of Infectious Disease and Molecular Medicine and the Department of Medicine, University of Cape Town, Rondebosch, Western Cape, South Africa
[4]Molecular Mycobacteriology Research Unit, University of Cape Town, Rondebosch, Western Cape, South Africa
[5]Medical Informatics, Berne University of Applied Sciences, Bern, Switzerland
[6]Paul G. Allen School of Computer Science & Engineering, University of Washington, Seattle, Washington, USA
[7]Centre for Infectious Disease Epidemiology & Research, School of Public Health & Family Medicine, University of Cape Town, Rondebosch, Western Cape, South Africa
[8]Population Health Sciences, Bristol Medical School, University of Bristol, Bristol, UK

**Acknowledgements** We would like to thank Mbali Mohlamonyane for setting up the bio-aerosol sample collection system, Zeenat Hoosen and Ronnett Seldon for developing the laboratory protocols. We are also grateful to the clinical manager,the staff members at the clinic and all patients who contributed to this study. Finally, we would like to thank the City of Cape Town for the support they gave us for the use of one of their clinic facilities. Research findings and recommendations do not represent an official view of the City of Cape Town.

**Contributors** Overall concept: KZ, CM, MB, RW, ME, LF. Medical informatics: SB, JR. Cough sound concept: XL, MS, CM, RW. Video sensor concept: KZ, JR, SB, ME, LF. Patient data concept: CM, KM, RW. Laboratory work concept: CM, ASK, KM, DW, RW. Modelling work concept: KZ, JR, SB, ME, LF. Pilot study coordination: CM, KM, RW. KZ, MB, ME, LF wrote the first draft of the paper, which was reviewed by all authors and revised on the basis of the comments received by co-authors. All authors approved the final version of the manuscript.

**Funding** The roll-out of this study is supported by a grant from the Swiss National Science Foundation (grant no. CRSK-3_190781). KZ and LF are supported by the National Institute of Allergy and Infectious Diseases (NIAID) through grant no. 5U01-AI069924-05, ME by special project funding (grant no. 174281) from the Swiss National Science Foundation.

**Competing interests** None declared.

**Patient and public involvement** Patients and/or the public were involved in the design, or conduct, or reporting, or dissemination plans of this research. Refer to the Methods section for further details.

**Patient consent for publication** Not required.

**Provenance and peer review** Not commissioned; externally peer reviewed.

**ORCID iDs**
Kathrin Zürcher http://orcid.org/0000-0002-7915-3194
Marie Ballif http://orcid.org/0000-0003-3133-3011
Anastasia Sideris Koch http://orcid.org/0000-0002-5897-4196
Keren Middelkoop http://orcid.org/0000-0001-9922-4263
Matthias Egger http://orcid.org/0000-0001-7462-5132
Lukas Fenner http://orcid.org/0000-0003-3309-4835

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
