## [Reviewer comments · BMJ Open]

ARTICLE DETAILS

TITLE (PROVISIONAL)	Novel approach to estimate tuberculosis transmission in primary care clinics in sub-Saharan Africa: protocol of a prospective study
AUTHORS	Zürcher, Kathrin; Morrow, Carl; Riou, Julien; Ballif, Marie; Koch, Anastasia; Bertschinger, Simon; Liu, Xin; Sharma, Manuja; Middelkoop, Keren; Warner, Digby; Wood, Robin; Egger, Matthias; Fenner, Lukas

VERSION 1 – REVIEW

REVIEWER	Meredith Brooks Harvard Medical School, USA
REVIEW RETURNED	07-Feb-2020

GENERAL COMMENTS	I would like to thank the authors for the opportunity to review their manuscript entitled “A novel prospective study design to estimate tuberculosis transmission in primary care clinics in African high-burden settings.” This study offers a novel approach with unique metrics to assess risk factors for airborne transmission at primary care clinics in high TB/HIV burden settings. This will allow for the modeling of the effect of potential interventions on reducing TB transmission in these settings. The results can guide plans for logistic- and infrastructure-based adaptations to improve the risk of TB transmission. While the manuscript is well written, I believe that it lacks some essential details for the reader to fully grasp the methods and how the data collected will be used. Some comments and suggestions follow. Overall, it would be beneficial to have study objectives very clearly laid out early on. It is very clear throughout that the study is designed to assess risk factors for TB transmission. However, it is only mentioned briefly and late that different types of intervention strategies, such as capping the number of individuals in the waiting room or having a separate area for coughers, will also be explored to assess how they may reduce the risk of TB transmission in these clinics. Additionally, only a few of these intervention strategies are mentioned. It would be preferred that the author’s lay out all strategies that will be explored so the reader can assess how impactful this research can be. Details are lacking in the methods section to allow readers to fully understand how data will go from collection to analysis. Specific comments on the methods below: Methods – Study Setup section: lacking information on the clinics in which this research will take place. More details regarding the number of clinics and what regions should be described.
---

	Methods – Patient-level variables – Video Sensor Data section: The person-tracker sensor system seems novel and incredibly useful for this type of research, however it is unclear exactly how it outputs data. Does someone have to watch the tape back and manual code or does it output code by patient? It would be useful to confirm how the measures that are not directly output by the data system are each going to be calculated—i.e., waiting times until medical consultation will be calculated as the duration from the entry of an individual into the clinic until they enter the TB treatment room. Similarly, how is the average distance between people recorded? The authors note that are data captured in the 0.25 second snapshots. Does each 0.25 second snapshot calculate the distance of each individual in the room in relation to each other individual in the room? Are these values then averaged over the entire day or period of tracking? Methods – Patient-level Variables – Patient Data: Should the first line of this section read “...clinical data from the patient CHART including date...” instead of just “...clinical data from the patient including date...”? Data are not being collected directly from the patients, correct? Please be more specific about the type of information that will be collected at the site-level. I.e., what types of infection control measures are being documented. Methods – Definitions: the definition of presumptive TB includes the review of clinical symptoms or signs suggestive of TB, but these type of data are not listed in the previous section as being extracted from the patient chart. Methods – Environmental Variables – CO2, Relative Humidity and Temperature Monitoring: The CO2 monitors are collecting such an important data component that their use should be described directly in this manuscript and not referred to as previously described. This protocol manuscript will serve as a reference to study-related details that you refer to when results are published so should contain full details instead of leading readers to multiple other references to search for details. The sentence “Humidity is associated with Mtb survival in the air” seems like it belongs in the introduction. In fact, rationale for why each of these measures are being used, such as the humidity statement, would be beneficial to readers because it is such a unique study design. Methods – Environmental Variables – Cough Monitoring: How will the frequency, intensity, and duration of coughing be reported? Purely by location/room per day? Or will these somehow be linked back to the movement data—such as how many individuals were in the room and how close were they together at the time of the most intense, longest duration coughing because that may be a time of greater risk of transmission as opposed to a time when only two individuals are in the room and are far apart. Methods – Environmental Variables – Bio-aerosol Sampling and Molecular Detection: The authors note that the dry filter units only
--	--

	collect air for 7 hours each day. Later on when describing the pilot study it is noted that that clinic was open from 7am-4pm (9 hours). Please describe when the filters were being used—during the busiest clinic hours; consecutive from opening? Was this standard or changed daily, as there might be great variability in when patients sick with TB are visiting the clinic. Here it is noted again in two places that a process was previously described elsewhere. Again, it is recommended that all details pertaining to this study protocol be contained here so that readers can refer here for all details once results are published. The methods section does not provide any information regarding how long data collection will take place – weeks or until a certain number of individuals have passed through? Methods – Statistical Analyses: The plan for how data will be used and linked is unclear. With so many different data points being collected at numerous points in time, it is unclear how the data will be aggregated over time periods, or whether they will be time-varying, for use in the model. To determine whether the models will be valid, the inputs need to be understood by readers. Also, will the relative humidity, temperature, cough sounds, etc be linked to moments in time so that a full description of each moment of time can be accounted for? This section is the first time that outdoor CO2 concentrations are mentioned. Will there also be a CO2 sensor outside of the clinic to act as a control? Methods – Mathematical Modelling: The authors refer to the mathematical model as Figure 3, but the actual figure itself is labeled as Figure 2. The hypothesis is stated in this section but would be beneficial to list earlier on when the objectives are laid out. How will the clinics be accounted for in the model—hierarchical model being used? More details about the actual methods for modeling should be described. The authors note that several metrics will be used to assess goodness of fit—please name the methods to be used. Also, please be clear about how the independent variables will be integrated and how a final model will be selected. The authors describe that factors will be removed from the models to see if certain interventions, such as limited the number of individuals in the waiting room, can lower the risk of transmission. What is considered an important reduction in risk—any reduction? A percentage? The gauge of a successful intervention should be documented. It would be useful to list all potential interventions that will be assessed. Right now it states “such as..” and then lists a few, implying that there will be more. Methods – Feasibility of the approach: pilot study: If limitations, such as stable power and wifi, were identified in the pilot study, will measures to account for these be implemented in this larger study? It is unclear whether these are actually planned for.
--	---

REVIEWER	Lina Davies Forsman Karolinska Institutet, Sweden
REVIEW RETURNED	09-Feb-2020

GENERAL COMMENTS	The authors are proposing a comprehensive study design in order to study TB transmission in primary care clinics in a high-endemic setting in sub-Saharan Africa. The manuscript is well written and in general adequately described. I have some minor suggestions for clarification. My main questions regards patient confidentiality and informed consent.  1. In the background section the drivers of the TB epidemic are described. For completeness, I would suggest acknowledging the influence of undetected and untreated DR-TB by adding that to the sentence. 2. How many clinics will be included in the study? How you made any power calculations? Since you have many measurements regarding exposure, you need sufficient amount of study participants as well as TB transmission (the outcome) in order to make robust statistical analyses. 3. Is there an age-limit to the patients? Will all patients be included? How will they be included? There if so mentioned of an informed consent. Since patient data will be collected it will be needed as well as information what happens if a patient does not want to participate. I found it a bit difficult to understand whether you will know which individual represents the “dot” in your person-tracker sensor system. “Anonymous data” will be collected, does that refer to the identity of the patient will be represented by a study number? If so, I would explain that separately and delete anonymous, as that alludes to that data will only be analysed on a group level. 4. Page 6 (numbering of the rows is inconsistent in this pate). Patient data “ We will extract routinely collected patient data from the patient....”. Don’t you mean from the medical records? Or will you perform interviews? Will the data be prospectively or retrospectively 5. Under mathematical modelling: “As we do not actually observe transmission events, we will use y_u, the number of Mtb genomes counted by the bio-aerosol sampling for each 6-hour period u, as the independent variable. “The structure of the sentence makes it difficult to follow, consider splitting it to two different sentences of make it clearer what the difference between y_u and u is. 6. In the discussion you clearly state that the implementation and evaluation of intervention is beyond the scope of this study. In the abstract however, the reader gets the impression that model will be used to estimate the effect of interventions. Consider clarify that the model may be used in future studies. 7. Consider adding a few sentences of the feasibility of the study by adding some of the results from the pilot study. It
--

	adds credibility that a pilot study has been performed. Refer to the figures and add some information if it was successful or not is needed. 8. Has the study been registered to clinicaltrials.gov? Please add if so. 9. I can't find any checklist attached (STROBE etc) 10. Figure 2 is difficult to understand. Consider changing Figure, if not add sufficient explanation as figure legends as to what you want to visualise and how the figure should be interpreted.
--	--

VERSION 1 – AUTHOR RESPONSE

Reviewer(s)' Comments to Author:

Reviewer: 1

Reviewer Name: Meredith Brooks

Institution and Country: Harvard Medical School, USA Please state any competing interests or state 'None declared': None declared

Please leave your comments for the authors below I would like to thank the authors for the opportunity to review their manuscript entitled “A novel prospective study design to estimate tuberculosis transmission in primary care clinics in African high-burden settings.” This study offers a novel approach with unique metrics to assess risk factors for airborne transmission at primary care clinics in high TB/HIV burden settings. This will allow for the modeling of the effect of potential interventions on reducing TB transmission in these settings. The results can guide plans for logistic- and infrastructure-based adaptations to improve the risk of TB transmission. While the manuscript is well written, I believe that it lacks some essential details for the reader to fully grasp the methods and how the data collected will be used. Some comments and suggestions follow.

Overall, it would be beneficial to have study objectives very clearly laid out early on. It is very clear throughout that the study is designed to assess risk factors for TB transmission. However, it is only mentioned briefly and late that different types of intervention strategies, such as capping the number of individuals in the waiting room or having a separate area for coughers, will also be explored to assess how they may reduce the risk of TB transmission in these clinics. Additionally, only a few of these intervention strategies are mentioned. It would be preferred that the author's lay out all strategies that will be explored so the reader can assess how impactful this research can be.

Details are lacking in the methods section to allow readers to fully understand how data will go from collection to analysis. Specific comments on the methods below:

Authors' response: Thank you for the careful and detailed review of our paper.

1. Methods – Study Setup section: lacking information on the clinics in which this research will take place. More details regarding the number of clinics and what regions should be described.

Authors' response: We added the following paragraph on the study setting and study population and procedure:

Study setting and study population

The study will take place at several ART or primary care clinics in countries participating in the International Epidemiology Database to Evaluate AIDS in Southern Africa (IeDEA-SA) collaboration with a high TB and HIV burden. These clinics are located in urban and peri-urban communities with predominantly young and black African residents. TB and HIV are both prevalent in these communities. (page: 5, line 144-149)

We also provide more information in the Ethics statement:

Ethics statement

The University of Cape Town Faculty of Health Sciences Human Research Ethics Committee (HREC/REF: 228/2019), the City of Cape Town (Project ID 8139) and the Ethics Committee of the Canton of Bern, Switzerland approved the pilot study. Expansion of the study will include other clinics in the greater Cape Town metropolitan area, but also clinics in Zambia, Zimbabwe or Malawi, including urban and rural sites to ensure representative and generalizable results. Separate ethical approval will be sought from the relevant ethics committees or institutional review boards. (page: 11, line 353-360)

2. Methods – Patient-level variables – Video Sensor Data section: The person-tracker sensor system seems novel and incredibly useful for this type of research, however it is unclear exactly how it outputs data. Does someone have to watch the tape back and manual code or does it output code by patient?

It would be useful to confirm how the measures that are not directly output by the data system are each going to be calculated—i.e., waiting times until medical consultation will be calculated as the duration from the entry of an individual into the clinic until they enter the TB treatment room. Similarly, how is the average distance between people recorded? The authors note that are data captured in the 0.25 second snapshots. Does each 0.25 second snapshot calculate the distance of each individual in the room in relation to each other individual in the room? Are these values then averaged over the entire day or period of tracking?

Authors' response: The raw data consist of the date, time of the day, the person's height and position (x-y coordinates) for each unique individual during the entire duration of their stay within the clinic. This data will be captured every 0.25 seconds. The raw data will then be parsed by a python script partially provided by Xovis. The python script will calculate the definitive persons' height, total movement, and observation time at different locations for each patient. Further, we will develop scripts for visualizing hotspots.

List of collected and computed data from the video sensors:

- Date, timeframe, and time point are provided by the tracking system (raw data)
- Person ID is provided by the tracking system, adjusted to start from 1
- x and y coordinates are provided by the tracking system
- Distance to next person is calculated based on the coordinates
- Persons' height is provided by tracking system
- Time spent at each different location: Number of minutes in which the individual was observed at each location in the clinic
- Total movement based on change in coordinates over time
- Total number of individuals overall or in a given zone (e.g. in the waiting room, at the registration or in the TB treatment room) every minute
- Total number of sitting individuals based on the height at a given timeframe every minute

The manuscript was revised as follows:

The raw data consist of the person's height, time, date, and the position (x-y coordinates) for each unique individual during the duration of their stay within the clinic. The data are captured every 0.25 seconds. The raw data are then parsed by a python script to calculate the height, total movement, and observation time at different locations, and to visualize hotspots. (page 6, line 160-166)

3. Methods – Patient-level Variables – Patient Data: Should the first line of this section read "...clinical data from the patient CHART including date..." instead of just "...clinical data from the patient including date..."? Data are not being collected directly from the patients, correct?

Authors' response: Yes, this is correct. We will not collect data directly from the patients or patient charts, but extract clinical data from the electronic registering system. We added the following sentence to clarify this:

We will extract clinical data from the electronic patient registry for all patients who visited the clinics during the four weeks of data collection. The data will include date and time of registration, age, sex, height (used for linkage with video sensor data), HIV status, presumptive TB (“TB suspects”), mode of TB diagnosis (sputum smear microscopy, Xpert MTB/RIF, mycobacterium culture), TB diagnostic test results, date of TB treatment start, current anti-TB treatment regimen. We will not extract any personal data, such as names or social security numbers. (page: 6, line 173-179)

4. Please be more specific about the type of information that will be collected at the site-level. I.e., what types of infection control measures are being documented.

Authors’ response: Thank you. We added the following sentence:

The clinic-level data will include information such as the setting (urban or rural), level of care, the number of patients seen each year, availability of adult and paediatric care, TB control measures (natural ventilation, use of masks, separation of TB patients or coughers), and the floor plan of the clinics. The data will be collected using the web-based REDCap data entry system (www.redcap.org). (page: 6, line 179-184)

5. Methods – Definitions: the definition of presumptive TB includes the review of clinical symptoms or signs suggestive of TB, but these type of data are not listed in the previous section as being extracted from the patient chart.

Authors’ response: Thank you for pointing this out. We will collect this information and revised the manuscript accordingly (see response to comment 3).

6. Methods – Environmental Variables – CO₂, Relative Humidity and Temperature Monitoring: The CO₂ monitors are collecting such an important data component that their use should be described directly in this manuscript and not referred to as previously described. This protocol manuscript will serve as a reference to study-related details that you refer to when results are published so should contain full details instead of leading readers to multiple other references to search for details.

Authors’ response: We added the following information to describe the CO₂, relative humidity and temperature monitoring in more detail.

We will use CO₂ monitors (Digital CO₂ Monitor Carbon Dioxide Meter XE-2000, XEAST, Guangdong, China) which include the COZIR-A sensor for ambient CO₂ levels of 0–1% (Gas Sensing Solutions, Cumbernauld, Scotland). The monitors will record indoor CO₂ concentrations (in parts per million [ppm]), temperature, and relative humidity (RH) at minute intervals. Data from the five monitors are stored and exported in serial digital format. At each clinic we will install five monitors to cover the most visited spaces and a control area outside the clinic (Figure 1). The monitors auto-calibrate over time to a standard minimum value of 400 ppm. This value is close to the average monthly outdoor carbon dioxide concentration measured at Cape Point, South Africa (405.48 ppm in December 2018). Monitors will be run for one week before the start of the study to allow them to calibrate and to confirm that they delivered comparable data. (page: 7, line 193-204)

7. The sentence “Humidity is associated with Mtb survival in the air” seems like it belongs in the introduction. In fact, rationale for why each of these measures are being used, such as the humidity statement, would be beneficial to readers because it is such a unique study design.

Authors’ response: Thank you for the comment. We moved this sentence to the Introduction. Also, we added a table (Table 1, see response to comment 11) showing the main data variables including the value/unit, a description and rationale for its inclusion in the study as well as the relevant reference.

8. Methods – Environmental Variables – Cough Monitoring: How will the frequency, intensity, and duration of coughing be reported? Purely by location/room per day? Or will these somehow be linked back to the movement data—such as how many individuals were in the room and how close were they together at the time of the most intense, longest duration coughing because that may be a time

of greater risk of transmission as opposed to a time when only two individuals are in the room and are far apart.

Authors' response: As assumed by the reviewer we will use the cough monitoring by location or room per time unit (minutes and day). Unfortunately, it is not possible to link the coughing to an individual or the movement data.

We clarified this in the manuscript as follows:

We will identify audio records with cough sounds and calculate the cough frequency, intensity and duration. The cough data will be linked with the CO2 data and video sensor data by time and date. Finally, we will probabilistically link the video sensor data (i.e. the moving dots) with the clinical data by the height of the patient, time and date of the visit. (page 7, line 213-217)"

9. Methods – Environmental Variables – Bio-aerosol Sampling and Molecular Detection: The authors note that the dry filter units only collect air for 7 hours each day. Later on when describing the pilot study it is noted that that clinic was open from 7am-4pm (9 hours). Please describe when the filters were being used—during the busiest clinic hours; consecutive from opening? Was this standard or changed daily, as there might be great variability in when patients sick with TB are visiting the clinic. Here it is noted again in two places that a process was previously described elsewhere. Again, it is recommended that all details pertaining to this study protocol be contained here so that readers can refer here for all details once results are published.

Authors' response: Thank you. We clarified this in the Methods section as follows:

Bio-aerosol sampling and molecular detection

We will collect Mtb particles from the air using mobile bio-aerosol sampling devices (Dry Filter Unit [DFU]-1000, Lockheed Martin Integrated Systems, Gaithersburg, MD, USA). The DFU 1000 is a portable biological air sampler. Ambient air is drawn through 1 µm polyester-felt filters at a rate of ~1000 l/min via an electrical blower. The DFU is a useful collection system for low concentration aerosols, capturing particles of 1 µm or larger, and allowing easy access for the retrieval of filters. One DFU will be placed in the waiting room and the other in the TB treatment room. Each DFU collects air for 7 hours (two periods of 3.5 hours) every day onto two filters. Due to logistical and human resource limitations, we will be able to change filters only twice per day. Therefore, we decided to run the mobile bio-aerosol sampling devices during the busiest time at the clinic, between 7 am and 2 pm. All other data will be collected from 7 am to 4 pm. (page 6-7, line 220-231)

In addition, we added the following text on the DFU to the same paragraph:

As previously described, 16 duplicate filters from each sampling session will be transferred to 50mL Falcon tubes and vortexed in sterile phosphate buffered saline (PBS) (with 0.05% Tween 80). Following centrifugation at 3750 rpm for 15 mins, filters will be removed and the pellet subjected to DNA extraction. DNA from Mtb cells will be extracted using an in-house lysis buffer with subsequent pelleting (centrifugation 13000 rpm for 10 mins) of DNA and resuspension in 50 µl of Tris-EDTA buffer (10mM Tris, 1mM EDTA, pH 8.0). Given that ddPCR is relatively robust against inhibitors, no further DNA purification will be required. The primer/probe combinations and reaction conditions for Mtb specific ddPCR have been described. Samples with known amounts of purified Mtb DNA (0.01ng and 0.001ng) will be included to serve as positive and nuclease-free water as negative control. The data generated from the ddPCR reaction will be analyzed via the Umbrella pipeline, using wells with a minimum of 10,000 droplets. (page 7, line 232-244)

10. The methods section does not provide any information regarding how long data collection will take place – weeks or until a certain number of individuals have passed through?

Authors' response: This has been clarified – see responses to comments above.

11. Methods – Statistical Analyses: The plan for how data will be used and linked is unclear. With so many different data points being collected at numerous points in time, it is unclear how the data will

be aggregated over time periods, or whether they will be time-varying, for use in the model. To determine whether the models will be valid, the inputs need to be understood by readers. Also, will the relative humidity, temperature, cough sounds, etc be linked to moments in time so that a full

Data source	Parameter	Description	Unit	Measurement taken by:
CO ₂ monitor	CO ₂	Observed CO ₂ concentration in the indoor air per minute and a control in the outdoor air. Based on CO ₂ levels (parts per million [ppm]) and people density, we will calculate rebreathed air volume (RAV), which is used as a proxy for airborne TB transmission. ¹	ppm	minute and date
	Relative humidity	Data on the effects of relative humidity on the survival of airborne bacteria are inconsistent. ² ³ However, a recent study found that relative humidity above 65% is associated with Mtb survival in the air. ⁴	%	minute and date
	Temperature	Temperatures above 24°C are required to reduce airborne bacteria survival. ^{2 3}	°C	minute and date
Cough recording	Frequency	One of the typical symptoms of TB is coughing; coughing is also the main way of transmission.	n	minute or day and date
	Duration	Duration of each cough is different from healthy and people with TB or other lung diseases. ⁵	sec	cough by minute and date
	Intensity	Intensity of each cough is different from healthy and people with TB or other lung diseases. ⁵	decibel	cough by minute and date
Mobile aerosol sampling	Mtb DNA copies	Detection of Mtb particles in the air by filter or per day (7am to 2pm). ⁶	Copies per microliters	filter (ca.3.5h sampling) or per day
Video sensor	Number of people	From the raw data (x-y coordinates) we can calculate the number of people at a given location by 0.25 seconds and by minute.	n of people	0.25 seconds or minute and by date
	Time spent at a given location	From the raw data (x-y coordinates) we can calculate for each person their time spent at different locations.	minutes	minute and date
Patient charts	Number of registered patients	All patients who are visiting the clinic are registered.	n of registered patient	minute and day
	Number of presumptive TB and of TB patients	From all registered patients we will know the number of presumptive TB and of TB patients.	n of presumptive TB and TB patients	minute and day

description of each moment of time can be accounted for?

Authors' response: We agree that the high number of collected data variables might be confusing. Therefore, we added a table giving an overview of the collected (Table 1, below). Please see also response to comment 16 below.

Table 1: Description of the measurements.

12. This section is the first time that outdoor CO₂ concentrations are mentioned. Will there also be a CO₂ sensor outside of the clinic to act as a control?

Authors' response: Yes, thank you. We will be using a CO₂ sensor outside of the clinic as a control. We added this information in the Methods section, where we explain the use of CO₂ monitors:

At each clinic we will install five monitors to cover the most visited spaces and a control area outside the clinic (Figure 1). (page 7, line 197-199)

13. Methods – Mathematical Modelling: The authors refer to the mathematical model as Figure 3, but the actual Figure itself is labeled as Figure 2.

Authors' response: Thank you. We corrected this error.

14. The hypothesis is stated in this section but would be beneficial to list earlier on when the objectives are laid out.

Authors' response: We included the hypotheses in the concluding paragraph of the Introduction:

We hypothesize that (i) exposure to *Mtb* particles at the clinic can be estimated by studying the patient flow and CO₂ levels; (ii) the number of individuals present, the rebreathed fraction and the frequency of coughs influences exposure to *Mtb*; (iii) clinics with small and crowded waiting rooms, low ventilation, and suboptimal patient separation have an increased risk of TB transmission. (page 5, line 129-133)

15. How will the clinics be accounted for in the model—hierarchical model being used? More details about the actual methods for modeling should be described.

Authors' response: We thank the reviewer for this remark. At the start, the number of clinics included is expected to be low, so data from each clinic will be modelled separately. If the number of clinics becomes larger than about 20 to 30, it will indeed become relevant to add a hierarchical structure to the model. However, we do not think we will be able to expand the study to this number of clinics soon due to the COVID-19 epidemic.

16. The authors note that several metrics will be used to assess goodness of fit—please name the methods to be used. Also, please be clear about how the independent variables will be integrated and how a final model will be selected.

Authors' response: We have clarified this as follows under the heading statistical analyses:

Statistical analyses

We will describe the data and examine associations between sources of data and then use results to parameterize a mathematical model (Table 1). The statistical analysis will quantify the joint association between the clinical and environmental variables and the number of *Mtb* particles measured by the mobile aerosol sampling. To this end, we will use Poisson regression, with the number of *Mtb* particles by the period of time as the dependent variable, considered as a proxy measure for the risk of TB transmission. As this variable is measured by periods of 3.5 hours, the other variables will be aggregated over the same periods. We will consider all combinations of independent variables, including second-order interactions, and compare the model using standard model selection methods. We will use variable selection methods in a Bayesian framework, including the deviance information criterion (DIC) and the leave-one-out information criterion (LOOIC). (page 8, line 247-258)

17. The authors describe that factors will be removed from the models to see if certain interventions, such as limited the number of individuals in the waiting room, can lower the risk of transmission. What is considered an important reduction in risk—any reduction? A percentage? The gauge of a successful intervention should be documented.

Authors' response: After having selected the model providing the best description of the number of *Mtb* particles measured by the mobile aerosol sampling (considered as a proxy for the risk of TB transmission), we will in a second stage use this model to simulate the effect of interventions. This will be done by fixing or modifying some parameters of the model. For instance, assuming that the rebreathed air fraction will be retained in the best fitting model, we could simulate the effect of regular ventilation of the waiting room by decreasing the rebreathed air fraction to some degree. We will then quantify the corresponding relative reduction in the number of *Mtb* particles as predicted by the

model. What will be considered an important reduction risk will depend on the size of the effect beyond sample variation, and the scale and difficulty of implementing the intervention.

18. It would be useful to list all potential interventions that will be assessed. Right now it states “such as..” and then lists a few, implying that there will be more.

Authors’ response: We explain this in the section on Mathematical modelling:

Having developed the model of the risk of TB transmission, we will evaluate the possible effect of interventions such as limiting the number of patients in the waiting room, separating coughers or increasing ventilation.

19. Methods – Feasibility of the approach: pilot study: If limitations, such as stable power and wifi, were identified in the pilot study, will measures to account for these be implemented in this larger study? It is unclear whether these are actually planned for.

Authors’ response: We have clarified this in the para on the Pilot study:

Data collection was successful overall, however, we experienced several power cuts during the pilot study. An important lesson learned is that power banks for laptops and WiFi routers are needed, as well as UPS power stabilizers and access to a generator as back-up for the DFU and CO2 monitors. (page 11, line 348-351)

Under Strengths and limitations we say:

Study limitations pertain to the need for stable electrical power and WiFi over 24 hours for data collection at the primary care clinic, which is often an issue in low- and middle-income countries. We will address this problem by using UPS power stabilizers with access to generator back-up power. (page 12, line 389-392)

Reviewer 2

Reviewer Name: Lina Davies Forsman

Institution and Country: Karolinska Institutet, Sweden Please state any competing interests or state 'None declared': None declared

Review comments

The authors are proposing a comprehensive study design in order to study TB transmission in primary care clinics in a high-endemic setting in sub-Saharan Africa. The manuscript is well written and in general adequately described. I have some minor suggestions for clarification. My main questions regards patient confidentiality and informed consent.

1. In the background section the drivers of the TB epidemic are described. For completeness, I would suggest acknowledging the influence of undetected and untreated DR-TB by adding that to the sentence.

Authors’ response: Thank you. We added references to drug-resistant TB to the following sentences in the Introduction:

Tuberculosis (TB), caused by the bacterium *Mycobacterium tuberculosis (Mtb)*, remains a major global public health problem, particularly in the context of HIV and drug resistance. (page 4, line 93-95)

Drivers of the TB epidemic in sub-Saharan Africa are HIV-infection and the resulting immunodeficiency (the strongest risk factor),¹ delayed diagnosis and treatment as well as undetected and untreated cases of TB or drug-resistant TB. (page 4, line 98-101)

2. How many clinics will be included in the study? How you made any power calculations? Since you have many measurements regarding exposure, you need sufficient amount of study participants as well as TB transmission (the outcome) in order to make robust statistical analyses.

Authors' response: Thank you.

Please note, that we will not actually observe transmission events. We will estimate the risk of transmission using a mathematical model which is based on a comprehensive set of data as well as published information. Please see also Table 1 as well as the newly added Table 2 which shows all data variables to be collected.

We did not consider that power calculation would be relevant at this preliminary stage: the proposed study is observational, novel in its context and measurements, and will require a complex model with multiple independent variables. With such a non-standard design, power calculations would require an in-depth simulation study, with many unknown parameters. We will consider conducting such a simulation study based on the pilot study, to put the preliminary findings in context and adapt future implementations. This will, however, be a major undertaking.

3. Is there an age-limit to the patients? Will all patients be included? How will they be included? There if so mentioned of an informed consent. Since patient data will be collected it will be needed as well as information what happens if a patient does not want to participate. I found it a bit difficult to understand whether you will know which individual represents the "dot" in your person-tracker sensor system. "Anonymous data" will be collected, does that refer to the identity of the patient will be represented by a study number? If so, I would explain that separately and delete anonymous, as that alludes to that data will only be analysed on a group level.

Authors' response: Thank you for the comment. We will strictly follow ethical and data protection standards. We will also seek ethical approvals both from the local authorities and the Swiss Cantonal Ethics Committee, as well as permission from the governmental authorities. See Ethics Statement.

We will not obtain individual, informed consent. Patient confidentiality is not a concern since patients will not be identified in the video data. Images on tracked individuals will not be stored and will not leave the sensor at any time. The sensors provide several types of data output streams, with four levels of privacy (see Figure below), which have to be selected when setting up a device. For our study, the level of privacy will be set to level 2. Its means no images can be exported and only a fully anonymized object stream (constant stream of the coordinates of moving dots) will be available. These moving dots will be probabilistically linked to the de-identified clinical data. We discussed these issues at length with the local Ethics Committee and the City Council in Cape Town in the context of the pilot study.

We have clarified this point in the section on video sensor data as follows:

The sensors have four levels of privacy. In our study, privacy will be set to level 2, which means that the data are a fully anonymized stream of the coordinates of moving dots. These dots will be linked probabilistically to the anonymized clinical data thus excluding any risk of re-identification. Further, the images of tracked individuals taken by the sensors will not be stored (see also the data privacy and security statement by Xovis). (page 6, line 165-170)

4. Page 6 (numbering of the rows is inconsistent in this page). Patient data “ We will extract routinely collected patient data from the patient....”. Don't you mean from the medical records? Or will you perform interviews? Will the data be prospectively or retrospectively

Authors' response: Please see response to comment 3 from reviewer 1. We will extract patient data from the electronic patient registry. (page 6, line 173-174)

5. Under mathematical modelling: “As we do not actually observe transmission events, we will use y_u , the number of *Mtb* genomes counted by the bio-aerosol sampling for each 6-hour period u , as the independent variable. “The structure of the sentence makes it difficult to follow, consider splitting it to two different sentences or make it clearer what the difference between y_u and u is.

Authors' response: Thank you. Done.

6. In the discussion you clearly state that the implementation and evaluation of intervention is beyond the scope of this study. In the abstract however, the reader gets the impression that model will be used to estimate the effect of interventions. Consider clarify that the model may be used in future studies.

Authors' response: At this stage, the main objective of this study is to demonstrate and quantify the link between environmental variables and the risk of TB transmission, taking the number of *Mtb* particles by period of time as a proxy. Once we obtain a model providing a good description of the risk of TB transmission as a function of several environmental variables, we will be able to simulate the effect of several interventions. For instance, assuming that the rebreathed air fraction will be retained in the best fitting model, we could simulate the effect of regular ventilation of the waiting room by decreasing the rebreathed air fraction. We will then quantify the corresponding relative reduction in the number of *Mtb* particles as predicted by the model. The actual implementation of these interventions is beyond the scope of this study, but we expect this study to give general directions about the expected effect of several interventions using simulations.

We have clarified this in the abstract as follows:

We will calculate rebreathed air volume based on people density and CO₂ levels and develop a mathematical model to estimate the risk of TB transmission. The mathematical model can then be used to estimate the effect of possible interventions such as separating patient flows or improving ventilation in reducing transmission. (page 2, line 61-65)

7. Consider adding a few sentences of the feasibility of the study by adding some of the results from the pilot study. It adds credibility that a pilot study has been performed. Refer to the figures and add some information if it was successful or not is needed.

Authors' response: We will publish the results of the pilot study in a separate communication. These analyses are still ongoing. We added the lessons learned from the pilot study as follows:

Data collection was successful overall, however, we experienced several power cuts during the pilot study. An important lesson learned is that power banks for laptops and WiFi routers are needed, as well as UPS power stabilizers and access to a generator as back-up for the DFU and CO₂ monitors. (page 11, line 348-351)

Please see also response to comment 18 from Reviewer 1.

8. Has the study been registered to clinicaltrials.gov? Please add if so.

Authors' response: This is not a clinical trial, and it, therefore, cannot be registered on clinicaltrials.gov.

9. I can't find any checklist attached (STROBE etc)

Authors' response: This is not a paper reporting the results from an observational study, therefore, no checklist is attached. But we agree that reporting guidelines are an essential tool for complete and adequate reporting. We added the following short paragraph on Reporting:

Reporting

The results from this study will be reported following the recommendations of the STROBE statement. (page 11, line 335-337)

10. Figure 2 is difficult to understand. Consider changing Figure, if not add sufficient explanation as figure legends as to what you want to visualise and how the Figure should be interpreted.

Authors' response: We explained the Figure in more detail as follows:

Output from video sensors with moving dots showing the tracked persons. The numbers in the dots indicate the height of patients. The different sectors covered by the sensors are merged. Images from the pilot study. (page 22, line 551-553)

VERSION 2 – REVIEW

REVIEWER	Meredith Brooks Harvard Medical School, USA
REVIEW RETURNED	06-Apr-2020
GENERAL COMMENTS	I thank the authors for the opportunity to re-read their revised manuscript. They were incredibly responsive to the initial review and have answered all questions and alleviated any concerns.
REVIEWER	Lina Davies Forsman Karolinska Institutet Sweden
REVIEW RETURNED	29-Mar-2020
GENERAL COMMENTS	Thanks for your thorough explanations, all my questions and misconceptions have now been answered. I'm surprised that informed consent was not needed for ethical approval since you are collected patient data (albeit anonymised) but of course that can vary from setting and country. Best of luck

VERSION 2 – AUTHOR RESPONSE

Reviewer(s)' Comments to Author:

Reviewer: 2

Reviewer Name: Lina Davies Forsman

Institution and Country: Karolinska Institutet Sweden

Please state any competing interests or state 'None declared': None declared

Please leave your comments for the authors below

Thanks for your thorough explanations, all my questions and misconceptions have now been answered. I'm surprised that informed consent was not needed for ethical approval since you are collected patient data (albeit anonymised) but of course that can vary from setting and country.
Best of luck

Lina Davies Forsman

Many thanks for reviewing our manuscript.

Reviewer: 1

Reviewer Name: Meredith Brooks

Institution and Country: Harvard Medical School, USA

Please state any competing interests or state 'None declared': None declared

Please leave your comments for the authors below

I thank the authors for the opportunity to re-read their revised manuscript. They were incredibly responsive to the initial review and have answered all questions and alleviated any concerns.

Many thanks for reviewing our manuscript.